# Barriers and Facilitators to Engagement in Collective Gardening Among Black African Immigrants in Alberta, Canada

**DOI:** 10.3390/ijerph22050789

**Published:** 2025-05-16

**Authors:** Destiny Otoadese, Issa Kamara, Elizabeth Onyango

**Affiliations:** 1School of Public Health, University of Alberta, Edmonton, AB T6G 1C9, Canada; otoadese@ualberta.ca; 2Sinkunia Community Development Organization, Edmonton, AB T5J 0L6, Canada; issa@sinkuniacommunity.org

**Keywords:** collective community gardens, immigrants, urban agriculture, cultural food security, barriers, facilitators

## Abstract

Background: Community gardens are increasingly popular in Canadian cities, serving as transformative spaces where immigrants can develop self-reliant strategies for accessing culturally familiar and healthy nutritious foods. However, numerous facilitators and barriers exist that limit the engagement of racialized groups such as Black-identifying immigrants. Using a socio-ecological framework, this research explores barriers and facilitators of engagement of Black African immigrants in collective community gardening in Alberta, Canada. Methods: The study adopted a community-based participatory research (CBPR) approach using mixed methods. Data collection included structured surveys (*n* = 119) to assess general engagement, facilitators, and barriers; in-depth interviews (*n* = 10) to explore lived experiences; and Afrocentric sharing circles (*n* = 2) to probe collective perspectives in relation to engagement in collective gardening. Participants were purposefully recruited through community networks within African immigrant-serving community organizations. Results: Our findings revealed how barriers at various levels of the socio-ecological model (SEM) interact to shape the interest and engagement of African immigrants in collective community gardening. Access to collective gardens was associated with significant benefits, including maintaining healthy foodways, knowledge exchange, growing social capital, and community connections that support overall wellbeing. Conclusions: This study contributes an accessible framework for understanding and addressing the complex barriers that limit engagement in community gardens for vulnerable communities, while highlighting opportunities for creating more inclusive and culturally responsive urban agriculture initiatives.

## 1. Introduction

Community gardens are an increasingly popular model of urban agriculture in cities in the Global North and are serving as transformative and restorative spaces for individuals and populations but more specifically for immigrants and refugees [1,2,3]. Different models of urban agriculture exist, including the traditional community gardens, backyard gardening, and collective community gardens [4,5]. For the purpose of this study, our focus is on collective community gardens. We use collective community gardens to refer to community gardens operated through shared ownership, collaborative labor, and communal harvest distribution. In these spaces, decisions about what to plant, how to maintain the garden, and how to distribute produce are made collectively by the community members [5]. This model and community gardens more generally have numerous benefits.

Recent evidence in migration and food security literature shows that engagement in collective community gardening creates opportunities for immigrants to reshape their relationships with the food system in host countries [1,2,6,7,8,9]. These gardening spaces provide self-reliance strategies for accessing healthy food options and producing culturally familiar foods [2,4,8,10,11,12]. Additionally, evidence has revealed that engagement in collective community gardening contributes to improved health and successful integration of immigrant and refugee communities [2,13,14]. This trend has been observed across cities where urban gardens have served as transformative spaces that bridge cultural divides while promoting food security and embodied wellbeing [15,16]. Beyond nutritional benefits, these spaces cultivate neighborly connections and the recreation of social supports for newcomers.

However, collective community gardens face several barriers including bureaucracies in land access, fragmented regulations, infrastructure limitations, and varying and/or competing priorities within communities and in responsible government departments [17]. For example, a recent study in the United States of America (USA) indicated that land access, including the physical location of the community gardens, is an important predictor of interest and participation [18]. Another study that focused on an exploration of experiences of South Asian refugees and immigrants found that barriers to participation in collective community gardening manifest in unique ways, including spatial and transportation challenges [1]. However, approaches that center the lived experiences of participants, especially immigrants, and how they perceive and use these spaces can inform the development of more inclusive collective community gardens [16,19].

Recent studies emerging from the Canadian context have recommended accessible green spaces, including collective community gardens, as important resources for mental health and wellbeing [11,20,21]. Unfamiliarity with the local environment and the cold Northern climate have been identified as barriers to engagement in community gardening in Canadian cities; yet, opportunities for obtaining healthy food, building new relationships, and emotional restoration remain key motivations that outweigh those barriers to participation [13,21]. Despite the known benefits of collective community gardening, especially for minoritized communities, immigrants in Canada continue to experience an array of barriers to engagement in urban agriculture [9,22]. For example, newcomers to Canada may have advanced farming knowledge in the context of their countries of origin but limited knowledge of gardening in Canada. Additionally, recent reviews of the existing literature have revealed that these individuals tend to experience downward socioeconomic mobility, requiring them to work multiple minimum-wage jobs to make ends meet coupled with other challenges, hence giving them limited time to engage in collective community gardening [22,23,24]. Based on our knowledge, limited to no primary research exists on the barriers to engagement in collective community gardening, particularly for Black-identifying immigrants, who have reported an increased risk of food insecurity [22,23,24].

Our study aimed to systematically address this knowledge gap by exploring the facilitators and barriers to engagement in collective community gardening. The focus of the study was on how gardening spaces can better serve to maintain healthy foodways post-migration and cultivate social connections that support the wellbeing and integration of African immigrants in Alberta, Canada. This study was guided by the following research question: What are the barriers and facilitators to engagement in collective community gardening among Black African immigrants settled in Alberta, Canada? The study was guided by a socio-ecological model to explore the individual, interpersonal, community, environmental, and systemic barriers and facilitators of engagement of immigrants.

### Theoretical Framework—Socio-Ecological Model

The socio-ecological framework [25,26] informed the conceptualization of this research, examining barriers and facilitators to engagement in collective community gardening. The model incorporates the interactive effects of individual, interpersonal, community, environmental, and structural factors to examine participation in gardening and to help understand the intersecting barriers operating at different spheres of influence [1,4]. The framework is particularly relevant for immigrant communities, as it will uncover how personal factors (like time and resource limitations) interact with interpersonal dynamics (such as cultural connections and discrimination) and broader structural constraints (including systemic inequities) to influence participation in collective community gardening [1]. At the individual level, we examine micro-level factors such as demographic characteristics, time constraints, gardening knowledge and skills, and economic circumstances. The interpersonal level encompasses social networks, cultural connections, experiences of discrimination, and relationship dynamics within garden spaces. Community-level factors include garden accessibility, decision-making processes, and organizational characteristics of collective community gardens. Environmental factors include climate constraints, growing seasons, and physical infrastructure, while structural factors examine systemic barriers like immigration policies, the economic environment, and resource allocation.

## 2. Materials and Methods

### 2.1. Research Design

We employed a community-based participatory research (CBPR) approach guided by principles of equitable partnership, co-learning, and community benefit [27,28]. The study was conducted in Edmonton, Alberta, a popular destination for immigrants of African descent in Canada [29]. Our community partner, Sinkunia Community Development Organization (SCDO), an African-led non-profit organization, played a crucial role in shaping research objectives and facilitating community engagement. Sinkunia CDO operates various settlement programs and provides culturally appropriate support services for African immigrants in Alberta, including initiatives like a collective community garden and partnership with the Edmonton Food Bank to improve access to culturally appropriate supports. A mixed methods approach was employed by using online surveys, in-depth interviews, and sharing circles. In this CBPR project, we maintained continuous engagement through regular meetings and feedback sessions with our community partner who was involved in the design, implementation, analysis, and knowledge mobilization [28,30].

### 2.2. Participant Recruitment

Participants were purposefully recruited in collaboration with SCDO and existing community networks, including the Somali Cultural Association, Kenyans in Alberta Association, Nigeria Association, and the Africa Centre. Participants were eligible if they met the following criteria: (1) recent immigrants (mostly within the past 10 years); (2) of African descent and identify as Black; (3) had lived in Edmonton for not less than 2 years; (4) aged 18 years or older; and (5) proficient in English and gave informed consent.

*Surveys:* The survey study (n = 119) instrument was designed to assess multiple levels of barriers and facilitators to engagement in collective community gardening. Additionally, the questionnaire assessed the general trends in the population regarding engagement in urban agriculture and the type of gardening. The questionnaire was administered through Survey Monkey, combining face-to-face interviews and telephone conversations to increase accessibility for the immigrants. The survey data were then exported into SPSS version 29 for analysis. Descriptive statistics, frequencies, and proportions were calculated to summarize the demographic characteristics of the sample and key variables.

*One-on-one interviews:* To obtain nuanced narratives, we conducted *n* = 10 in-depth interviews (IDIs) with purposefully selected participants from the survey sample. Participants were selected to ensure adequate representation, considering factors such as age, gender, ethnicity, and year of arrival. The interviews were designed as semi-structured, open-ended conversations, creating a “listening and conversational space” [31] where participants could freely articulate their experiences, views, and narratives. Interviews lasted approximately 35–60 min each and were conducted virtually on Zoom to accommodate participant schedules and preferences.

*Afrocentric Sharing Circles:* Participants were engaged in facilitated online sharing circles (*n* = 2). In these sessions, we designed the process to create a safe space and encourage participants to engage authentically as Black people and to share their stories in a non-judgmental and culturally affirming space. This approach created space for dialogue and allowed us to probe for deeper insight where necessary. Participants shared personal and collective experiences of facilitators and barriers to engagement and general experiences with collective community gardening in Edmonton, Alberta.

All IDIs and Afrocentric sharing circles were digitally recorded, transcribed verbatim, and imported into NVivo version 15. Transcripts were then iteratively analyzed deductively (using the socio-ecological framework) and inductively to allow new themes and sub-themes to emerge and to develop a standard set of codes describing patterns observed in the data. The coding process followed Braun and Clarke’s [32] six steps for thematic analysis: (1) data familiarization; (2) initial code generation; (3) theme identification; (4) theme review; (5) theme definition and naming; and (6) report production. The coding process was iterative, with line-by-line content analysis. We integrated qualitative narratives with survey data to strengthen the validity and depth of our findings. This triangulation approach enabled a comprehensive analysis of participant experiences while preserving the richness of individual accounts.

### 2.3. Ethics and Participant Protection

This study received approval from the University of Alberta Research Ethics Review Board (Protocol ID: Pro00134979). After receiving information about study objectives, benefits, and potential risks, participants provided informed consent (written or verbal). With permission, virtual interviews were recorded. To ensure confidentiality, we collected no personal identifiers, and participants had complete autonomy to decide on the details of information they wished to share. Participation was voluntary, and participants had the right to withdraw at any time. Participants each received honoraria of CAD 25 for survey completion and CAD 35 for in-depth interviews or Afrocentric sharing circles to acknowledge their time and contributions to the study. All data were de-identified and securely stored following the University of Alberta data management guidelines.

## 3. Results

In this section, we present the demographic information of the study participants and the common types of community gardens reported by the participants. Subsequently, this is followed by a synthesis of the facilitators and barriers to engagement in collective community gardening

### 3.1. Socio-Demographic Characteristics of Survey Participants

Table 1 presents the demographic characteristics of survey study participants. More than half of the participants (55%) were female, and most of them engaged in backyard/home gardening (41.2%) and collective community gardening (37.0%). On average, the participants were relatively young and had a median age of 39 years, indicating additional responsibilities with significant time and resources dedicated to childcare. Furthermore, most participants had migrated within the last 10 years with at least 7 out of 10 participants having post-secondary education, spanning college qualifications to a post-graduate degree. However, only about half were formally employed. One in five had more than one job, indicating that such highly educated newcomer individuals, still navigating socioeconomic and cultural changes, are taking up multiple low-wage jobs to make ends meet.

### 3.2. Facilitators of Engagement in Collective Community Gardening

Survey findings showed that most immigrants engage in collective community gardening to gain community connections and to build new relationships. Other notable facilitators included having access to land, knowledge, and resources, as well as being supported by community organizations (see Figure 1 below).

Analysis and synthesis of qualitative data revealed the individual-, interpersonal-, community-, and environmental-level facilitators of engagement. At the individual level, immigrants highlighted (a) connections to agriculture and food production; and (b) the associated health and wellbeing benefits as potential motivators. The interpersonal- and community-level facilitators were (c) the social connections and opportunity to meet new people coupled with the existence of the necessary resources such as land and established community gardens in neighborhoods.

Connection to agriculture and cultural foods: It was observed that for some participants, a perceived connection to growing food was a key motivation. This was notable among participants who expressed how agriculture was a part of their identity in their home countries before migration. Therefore, engaging in collective community gardening provided a sense of continuity and place making through the maintenance of traditional foodways post-migration. Participants expressed strong interest in growing culturally familiar vegetables and herbs, viewing collective community gardens as potential spaces for cultural food sovereignty.


*I have been involved in collective community gardening. I love gardening a lot, unfortunately, I didn’t know about it earlier. It is one of the best things one can ever do. Any community or any culture that treasures farming, they can thrive in these spaces. They don’t need to depend on external help for survival. But if people become lazy, and they think that everything should come from their government and sit down and wait for the government to do something, then it’s a recipe for disaster, for the people, or for the community. So yes, I have been involved, and I would love to continue.*
(IDI-06)


*I can say it’s important to me, partly because these are the foods that I have grown up with, and also because I understand the nutritional value of some of these foods, especially the vegetables.*
(IDI-02)

***Perceived health benefits of engagement in collective community gardening:*** Participants expressed multiple perceived benefits of and motivation for engaging in collective community gardening, including access to fresh organic vegetables, physical activity, and social supports. Gardening spaces create opportunities for people to have a sense of belonging by being part of a community activity that not only enhances the health and wellbeing of individuals but also those of the community at large.


*Collective community gardening comes with a lot of benefits. To start with, you meet new people, people from different backgrounds and you form a community. We have different communities that we belong to. But you become part of a community, another community that is different from the ones that you’re familiar with.*
(IDI-05)

***Meeting people, forming connections, and building social capital in the community:*** This is underscored by participants who perceived collective community gardens as spaces where they can come out to meet people, cultivate new relationships, and nurture support for overall wellbeing and intercultural learning. Social capital played a role in engagement, with participants who knew someone already involved in collective community gardening expressing greater interest.


*Because I once saw, my friend bring tomatoes from a collective community garden, not the one from the shop. From a real garden. It made me feel like, “Wow, is this being done here?” I was wondering how people were able to plant these things. So that got me curious that if there is a way, then maybe I will try it and learn from it.*
(IDI-04)

Building relationships within the community emerged as another motivation; collective community gardens are seen as welcoming spaces for knowledge exchange between and amongst community members and across generations.


*In the collective community garden, because we have people from different countries, they come, and they’re like, “What are you growing there?” Vegetables that they’ve not seen, or they are like, “How do you prepare this?” Like when they see me grow a lot of kale, then they’re like, “What are you going to do with all that?”, then with that, we now start talking about, you know, how I use it, how I’ll prepare and store it. And yeah, so the collective community garden, it’s helped me learn quite a lot, meet people, and see how they are planting, you know, different ways of doing that because I learn something new.*
(IDI-02)

When gardens were accessible and culturally welcoming, they served as important and valued community spaces. Opportunities to meet other people who shared the same experiences in the collective community garden were identified as a key facilitator; cultivating such community and neighborly connections created an enabling environment where individuals felt welcome, appreciated, and valued.

### 3.3. Barriers to Engagement in Collective Community Gardening

Our analysis of both the qualitative and survey data revealed five interrelated themes. Guided by the socio-ecological framework that characterizes barriers across multiple levels, we categorized the themes into the following: (1) personal/individual-level factors, (2) interpersonal/social factors, (3) community-level/organizational factors, (4) environmental factors, and (5) structural/systemic barriers.

Based on the survey data, Figure 2 shows that personal/individual-level factors such as not having knowledge or access to information about collective community gardening initiatives, language barriers, and busy schedules were common barriers. Others report distance from the community garden, lack of reliable means of transportation, and lack of land as some of the community-level factors. At the interpersonal and structural levels, discrimination and limited access to land were some key barriers identified by the immigrants. Environmental factors, particularly extreme weather conditions such as long and cold winters coupled with wildfires during the summers, were some possible barriers to engagement in collective community gardening for the immigrant communities. Our synthesis of the qualitative data generated similar and additional sub-themes as reflected in Figure 3. Figure 3 illustrates the barriers at the different levels, and each level is accompanied by specific factors: individual factors (e.g., busy schedules, financial pressures), interpersonal factors (e.g., community connections, racial discrimination), community factors (e.g., accessibility, decision-making), and environmental factors (e.g., climate, growing seasons). In the subsequent section, we discuss the different barriers in detail and share verbatim quotes from the qualitative data.

#### 3.3.1. Personal or Individual Level

The demands of managing multiple responsibilities, particularly among those serving as primary breadwinners or caregivers, create time and resource constraints. Most participants worked multiple jobs and had childcare responsibilities while also attending school. One participant noted the following in the interviews:


*For those of us that are immigrants, one major barrier will be the time factor…. You see, many families, especially in the Black community, are working, trying to survive, and many of us have young ones, because we are immigrants from not more than 20 years. Let’s say from 15 years up, so many of us have new families, young children, so time would be a factor, the time we go to work, the time we go for their businesses. It could be a factor for our people to come out.*
(IDI-03)

Financial struggle and childcare responsibilities further compounded these constraints, as mentioned by most of the participants.


*You know, people are thinking about how to pay their rent, how to pay their bills. So that can be one barrier, why people are not able to go and volunteer to do collective community gardening, because they are thinking about their bills, you know? Like for me, I work twelve hours. You work twelve hours. When you come home, you have to do a lot of work again, cook, take care of the kids, so sometimes it’s challenging. It’s a big challenge.*
(IDI-01)

Majority of the participants reported a lack of knowledge or information about collective gardening opportunities as a barrier.


*I haven’t heard of community gardening actually, to be honest. I haven’t and that was probably why I as a person has not participated. But I mean, I would love to. I would love to see how that would work. I’m open to trying new things, and something I think it’s available. Of course, but why I haven’t done it is because I have never had an opportunity. In fact, I’m just hearing about it for the first time.*
(P-01 SC-2)

This information gap was mostly among newcomers, who expressed uncertainty about where and how to access collective community gardens. Language barriers, reported by some participants, complicated access to information and participation.


*Like I said, I had no idea that a collective community garden exists in Edmonton, but if I had known that there was a community garden, I would have wanted to be part of it. But what would have hindered me from joining such initiatives would probably be my current engagement, which is my studies. I think my studies would be the one of the major factors I would think that would hinder me from being a part of the community. But as someone who is a garden person, I would definitely make out time.*
(IDI-07)

In addition to individual-level factors, our analysis also uncovered another sphere of influence relating to interpersonal relationships and networks of community connections.

#### 3.3.2. Interpersonal and Social Dynamics

Interpersonal and social dynamics can create barriers that may be external to the individual but can influence interest and engagement. Some participants recalled experiences of othering and racial discrimination.


*My friend and I are the only Black people in that group. I don’t mind saying this, sometimes we’ve faced racial discrimination. Not from everybody. No. But a few people, some people. There are people who love us for who we are, no matter our color. Such things happen. We face discrimination in the area of work. Sometimes you will see it, not all the leaders. But some of them, when we come, they will want us to do the bigger work, the work that requires more energy, but during harvesting we will be left out. Or others, are allowed to harvest whatever they want, but whenever it comes to people of color like us, we are treated differently. But we still continue to go, to attend. That is where we now wish, oh, [pause] how we wish we have our own community garden.*
(IDI-03)

In situations where participants felt like they were treated unfairly, or where they felt unwelcome, such individuals can be demotivated, creating interpersonal barriers to engagement. Our analysis further uncovered community-level factors that operate outside of personal and interpersonal factors to create patterns of inclusion or exclusion.

#### 3.3.3. Community-Level and Organizational Factors

Non-accessibility emerged as another barrier; over a third of survey respondents faced distance and transportation challenges.


*I know some people who are not able to come to the collective community gardens because they are far from where they live. Like where we farm, for instance, I think it’s maybe more than 20 km away from where we live, so it depends on, do people have time to go to the community garden, and how close is it?*
(IDI-02)


*You know, people can even have spaces that are closer to where they live. People can decide that, “Oh, I would belong to the one that is closer to my house,” So it would be more convenient. Besides creating awareness, convenience is also another thing. My husband, for instance, has never come to the collective community garden. Because we live in the Southwest, and the collective community garden is on the Northside. My husband does not like going far. And even when we offer to take him, he’s like, “No, I’m not going.” So some people would not want to maybe drive ten minutes, 15 min, you know, between ten and 20 min to get to the collective community garden. So convenience is a barrier for some people.*
(IDI-05)

Non-participatory decision-making regarding what should be grown was identified as another barrier:


*There is a time I had wanted to join a certain collective community garden. But then I was like, “When it comes to the collective, how do people make decisions on the type of vegetables or fruits to grow?” “How do you decide, or how do you agree that we are going to grow this type of vegetable and not this other one?”. So collective community gardens, it really depends on who is involved, because we have different tastes, different desires. In a collective garden, how do people make decisions on what to grow?*
(IDI-02)


*I am involved with a collective community garden, here in Edmonton, but you find that all we do is to garden Canadian food, or even other foreign foods, and not African, but European foods. Because I remember we planted lots of garlic. They bring different species of garlic from Germany, Bulgaria, although they will say, “This one is German, this Bulgarian’s flavour,” all those species of garlic. And others, maybe like tomatoes, different types of tomatoes, they have the Italian, they have the Mexican, and things like that, but nothing from Africa.*
(IDI-03)

In addition, participants identified environmental constraints to collective community gardening in Edmonton’s Northern climate.

#### 3.3.4. Environmental Constraints

The cold weather and short growing season characteristic of Canadian weather posed specific challenges for growing culturally familiar tropical vegetables that typically require longer growing seasons. One IDI participant narrated the following:


*Yeah, so I grow my own vegetables, I only buy vegetables when I’ve not been able to grow enough. Like last year I wasn’t around. I was visiting home, so we did not grow a lot of vegetables here which means that now we are buying. So I can’t wait for May to grow my vegetables. The growing season is short, from May to mid-September. Yes, if your plants go to October, you are taking a risk, because frost can come anytime, especially at night. So normally sukuma wiki [kale] can survive into maybe mid-October, but the other vegetables, once they are hit with the first frost, they just die. So again, when you are planting vegetables, you have to know what you must harvest by September and know what you can risk leaving in the garden.*
(IDI-02)

Climate change impacts and unpredictable growing cycles were noted as emerging concerns creating an uncertain environment for gardeners.


*As immigrants, we are not able to afford a greenhouse unless we have to rent a place in the greenhouse, which is expensive. We have access to shared community spaces. But, you know, depending on the season, if there is rain, then you will get a good harvest there. If there’s no rain, you will not get any harvest, and if the rain is too much, it destroys away your harvest [Laugh]—like last year we planted our things in one of the collective community garden, and you know, there was a heavy storm, hailstorms that destroyed everything that we planted there. It was a big loss. [Laughs] Yeah, so that brings another distress for the collective community garden. So the greenhouse, it’s necessary, so that if there is a big storm, or whatever, hailstorm, it won’t be affected.*
(P01 SC 1)

#### 3.3.5. Structural Barriers

Systemic racism and racial discrimination, reported by some participants, manifested in both subtle and overt ways, affecting feelings of belonging and inclusion in collective community garden spaces. Devaluation of education certificates and previous work experience, rising costs of living, unemployment or underemployment, and high housing costs have created financial pressures that limit the capacity of immigrant to engage in community gardening. Import restrictions on inputs such as seeds from participants’ countries of origin significantly affected their ability to grow and experiment with some traditional varieties and herbs. The influence of these structural barriers created compounded challenges for participants navigating different barriers to engagement.

## 4. Discussion

This study has generated novel insights into the complex interplay of the factors influencing Black African immigrants’ engagement in collective community gardening. The facilitators identified in this study illustrate the multi-functional benefits of collective community gardening for immigrant communities. A strong connection to agriculture and food production, often rooted in pre-migration experiences, is a primary facilitator. Most immigrants have prior connections to agriculture and more specifically, to the land, which then acts as a motivation for their engagement in gardening [9,18,20,33]. These findings are consistent with previous studies that have demonstrated that most immigrants are motivated by the desire to maintain their traditional foodways and to reconstruct their social connection through engagement on the garden [12,34,35].

Additionally, the perceived health benefits including access to fresh produce, physical activity, and improved mental wellbeing align with previous research on the role of community gardens in embodied wellbeing and social integration of immigrants [2,3]. Engagement in collective community gardening created opportunities for individuals in this study to experiment with traditional crop and vegetable varieties, maintain healthy food cultures, and build social capital through societal connections. Such social networks also support newcomer resettlement through meeting new people, learning about Canadian culture, and developing context-specific agricultural skills. These findings suggest that local governments and immigrant-serving organizations should incorporate culturally syntonic initiatives such as collective community gardens into their programming; policymakers have the responsibility to expand growing spaces in immigrant-dense residential areas.

Evidence from the current study also shows that collective community gardens are helpful community resources. Participants who were actively engaged in collective gardening often did so through existing community connections. Meeting people who share similar experiences and backgrounds emerged as a key motivation. Our analysis reveals two roles collective community gardens play: building social capital and knowledge exchange. Social capital facilitates initial access and supports knowledge transfer about growing food in Canada. These findings extend previous work by Kingsley et al. [36] on the role of social networks in immigrant food systems by demonstrating how these relationships play out in spaces of urban agriculture.

However, limited access to land, irrigation facilities, and other essential inputs may constrain people’s interest in engaging. At the individual level, time constraints emerge as a significant challenge, reflecting the broader challenges of immigrant families balancing multiple jobs and responsibilities. Most immigrants have multiple pressures competing for limited time and resources. The emphasis on financial pressures and childcare responsibilities reveals how individual barriers are often manifestations of broader structural inequity. Economic pressures create systemic barriers influencing such individuals’ ability to engage in community-driven initiatives. The fact that most participants lacked information about collective community gardening opportunities suggests current efforts may not effectively reach some immigrant communities, particularly recent arrivals. Language barriers reported by some participants are a significant barrier to resource navigation and accessibility.

Individual barriers also intersect with interpersonal dynamics in collective community gardens, where experiences of discrimination and cultural exclusion can create additional barriers to engagement in collective community gardening. This finding adds nuance to the existing literature on community gardens as sites of social integration [1,2,6,7,8,35], suggesting that without intentional attention to cultural sensitivity and inclusion, these spaces may inadvertently reproduce broader social and structural patterns of inequity. Furthermore, these findings highlight the need for flexible collective community garden programming that accommodates diverse work schedules and family commitments of immigrant participants. Immigrant-serving organizations should consider outreach strategies through trusted community networks to address information gaps, in addition to offering multilingual resources and childcare support during garden activities. Future research should examine how successful garden initiatives navigate these multilayered barriers and identify effective strategies for creating more accessible and welcoming collective garden spaces for diverse immigrant communities.

Analysis of community-level factors reveals how representation, accessibility, and inclusive decision-making concerns shape interest and engagement in collective community gardening. The organizational structure of collective gardens significantly influences participation. Non-participatory decision-making can create barriers to meaningful engagement. Such models may exclude certain perspectives, whereas inclusive decision-making processes that involve participants in determining what and how crops are grown can create welcoming spaces that reflect the diversity of experiences, traditions, and perspectives that enrich collective community gardens. These findings suggest the need for regular opportunities for the gardening participants to provide feedback on structure and activities, with transparent processes for implementing community suggestions.

In addition, findings from this study raise important questions about the viability of collective community gardening as a strategy for addressing cultural food insecurity in Northern climates of Canadian Prairie Provinces such as Alberta. Edmonton’s short growing season (May to mid-September) creates significant challenges for cultivating tropical vegetables that typically require longer. These environmental barriers are compounded by limited access to protected growing spaces like greenhouses, making year-round production of culturally familiar vegetables nearly unfeasible. Climate change further exacerbates these challenges, with increasing weather unpredictability particularly affecting those without access to protected growing spaces [37]. However, investment in infrastructure such as community greenhouses could transform these spaces from seasonal projects into sustainable, year-round resources for food production, community building, and neighborhood revitalization.

Our findings reveal how systemic racism operates as a structural barrier affecting multiple aspects of participation in the gardening activities. This manifests in both subtle and overt ways, from the distribution of garden infrastructure to decision-making processes about what should be grown. Additionally, economic structures create pervasive barriers to participation in gardening. Despite high educational attainment among immigrants, many face significant employment challenges that directly affect their ability to engage in community gardening. In the case of community gardening, systemic racism could also influence resource allocation, land use priorities, program design, and distribution of gardening infrastructure. Community gardens in Edmonton, like most cities in North America, show a pattern of concentration in predominantly non-immigrant neighborhoods, reflecting and reinforcing existing spatial inequities [12,16,23,38]. Furthermore, funding models and program requirements may downplay the needs and circumstances of immigrant communities, therefore creating additional barriers to meaningful engagement.

While community gardens are resources for community and neighborhood revitalization, the current structural environment creates pervasive barriers that exclude the communities that could benefit the most from collective community gardens. While improved information sharing and culturally appropriate programming are important, meaningful change requires addressing the underlying systemic inequities that create barriers to participation. This should include policy changes around seed importation, investment in greenhouse infrastructure in immigrant neighborhoods, reform of land use regulations, and strengthening ongoing efforts to address economic and racial inequity.

## 5. Limitations

Recruitment through community organizations may have underrepresented or excluded recent or certain groups of immigrants. However, previous studies have adopted similar recruitment strategies, and this approach enabled access to participants from 13 African countries, providing diverse perspectives. Second, while findings from Edmonton may not generalize to other Canadian cities, our socio-ecological model (SEM) offers an accessible template for examining similar dynamics in other cities. Third, although focusing only on Black African immigrants excluded other immigrant groups, this targeted approach revealed important insights about sociocultural, and structural barriers faced by this growing yet understudied population.

## 6. Conclusions

In conclusion, this study makes significant contributions to the literature and practice. The study highlights the relevance of collective community gardening in addressing social isolation, cultural food insecurity, and supporting mental health and wellbeing. However, a number of facilitators and barriers influence the participation of immigrants in collective community gardens. The facilitators and barriers operate at different levels—individual, interpersonal, community, environmental, and structural spheres. Cutting across the various levels are factors such as connection to agriculture and food production, the associated health and wellbeing benefits such as access to fresh produce, and the opportunities to build new connections as some possible motivations for immigrants. With reference to the barriers, busy schedules emerged as a common individual barrier, which reflects broader structural challenges around employment precarity and economic pressures facing immigrant communities. Our findings highlight how social capital operates as both a barrier and facilitator, with community connections simultaneously enabling garden access for some while reinforcing exclusion for others. Extreme climate conditions, coupled with increasingly unpredictable weather, are some critical barriers to engagement in urban gardening. Investment in community greenhouse infrastructure could extend the short growing seasons, protecting vegetables from increasingly volatile weather and enabling year-round participation. Such infrastructure could transform collective community gardens from seasonal projects into sustainable, year-round spaces for food production, community building, and neighborhood revitalization. As Canadian cities become increasingly diverse, equitable access to collective community gardens is crucial for supporting immigrant wellbeing and social integration. This study contributes an accessible framework for understanding and addressing the complex barriers that can limit access for vulnerable communities, while highlighting opportunities for creating more inclusive and culturally responsive urban agriculture initiatives.

## Figures and Tables

**Figure 1 ijerph-22-00789-f001:**
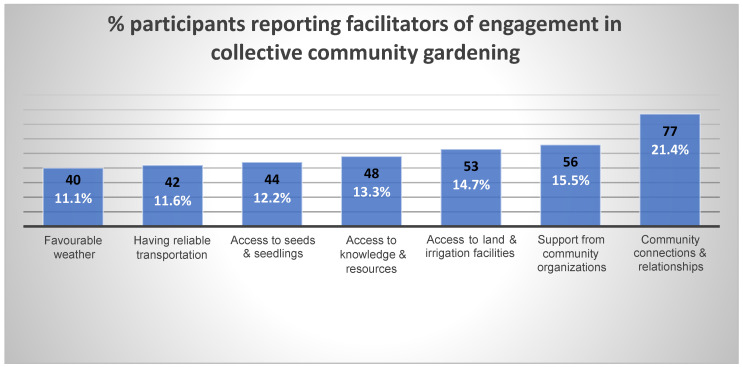
Proportion of participants reporting facilitators to engagement in collective community gardening.

**Figure 2 ijerph-22-00789-f002:**
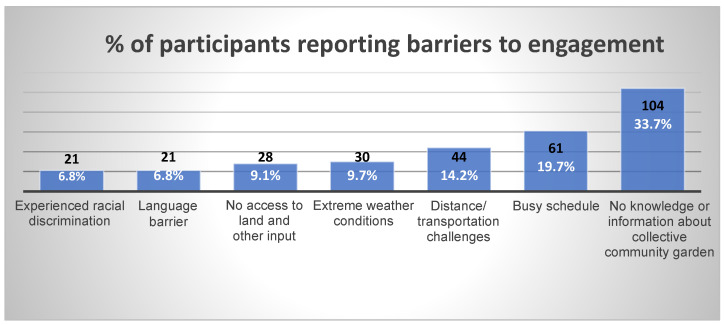
Proportion of participants reporting barriers to engagement in collective community gardening.

**Figure 3 ijerph-22-00789-f003:**
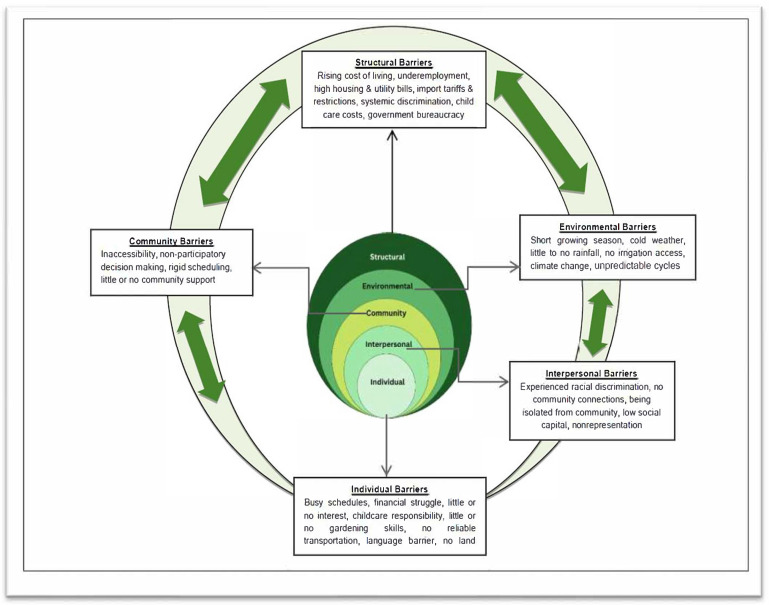
Conceptual model for analyzing barriers to engagement in collective community gardening.

**Table 1 ijerph-22-00789-t001:** Participant demographics of online survey.

Survey Participant Demographics, n = 119	
Gender, n (%)	
Female	65 (54.6)
Male	54 (45.4)
Year Born, median (range)	1986 (1948–2006)
Year Arrived in Canada, median (range)	2014 (1978–2024)
Employed, n (%)	
Yes	66 (59.4)
No	45 (40.6)
If yes, do you have more than one job? n (%)	
Yes	13 (19.7)
No	53 (80.3)
**Highest Level of Education, n (%)**	
No formal education	1 (0.8)
Elementary/primary complete	3 (2.5)
High school	29 (24.4)
Post-secondary	86 (72.2)
**Marital status, n (%)**	
Never married (single)	50 (44.2)
Married	56 (49.5)
Common law	2 (1.7)
Separated/widowed	5 (4.4)
**Household size, n (%)**	
Single person	11 (9.2)
2–5 people	43 (36.1)
More than 5	65 (54.6)
**Household Structure, n (%)**	
Female-centered	28 (23.5)
Male-centered	21 (17.6)
Nuclear	54 (45.4)
Extended	16 (13.4)
**Common types of gardening, (%) ***	
Home/backyard gardening	49 (41.2)
Collective community gardens	44 (37.0)
Community gardens	33 (27.7)
No identified	22 (18.5)

* Participants were allowed to select more than one response option.

## Data Availability

Data supporting the results reported in this study are hosted on the University of Alberta Google Drive and are not publicly available currently but are available on request.

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
