# Peer review of "Barriers and Facilitators to Engagement in Collective Gardening Among Black African Immigrants in Alberta, Canada"

_ijerph, 2025, doi:10.3390/ijerph22050789_

Round 1
Reviewer 1 Report
Comments and Suggestions for Authors
See attached file

See attached file
Author Response
Reviewer #2
Barriers and Facilitators
I enjoyed reading this article, and am heartened to see a study on community building through foodways – something that is close to my heart and I see as a vital means of repairing the fissures in our world. Canada is a fast-diversifying country with immigrants from all over the world, making study like this necessary to aid in the development of a more well-integrated, pluralistic society.
However, this article has a lot of problems both grammatical and conceptual. As detailed below, I found several parts of the manuscript to be either confusing, out of balance, or ill- defined. There are an abundance of concepts to the point where many are left to atrophy in single passages and not worked into a larger picture. The manuscript will require extensive re-writing for clarity, both in writing style and for what the major point it is trying to convey.
Section by section considerations:
Methods – Which of the authors contributed in each of the data gathering endeavors? Also, were the questions for the interviews determined at the same time as the survey, or in consideration for the results of the survey? While this is addressed in the Author Contributions, it is important to distinguish methodologically how diferent sources of data were obtained.
We have extensively revised the methods section and incorporated the suggestions. The methods section now reads as follows:
- Materials and Methods
Research Design
We employed a community-based participatory research (CBPR) approach guided by principles of equitable partnership, co-learning, and community benefit (Israel et al., 1998; Minkler & Wallerstein, 2008). The study was conducted in Edmonton, Alberta, a popular destination for immigrants of African descent in Canada (Statistics Canada, 2021). Our community partner, Sinkunia Community Development Organization (SCDO), an African-led non-profit organization, played a crucial role in shaping research objectives and facilitating community engagement. Sinkunia CDO operates various settlement programs and provides culturally appropriate support services for African immigrants in Alberta, including initiatives like a collective community garden and partnership with the Edmonton Food Bank to improve access to culturally appropriate supports. A mixed methods approach was employed by using online survey, in- depth interviews and sharing circles. In this CBPR project, we maintained continuous engagement through regular meetings and feedback sessions with our community partner who was involved in the design, implementation, analysis, and knowledge mobilization (Lincoln & Guba, 1985; Minkler & Wallerstein, 2008).
Participant Recruitment
Participants were purposefully recruited in collaboration with SCDO and existing community networks, including the Somali Cultural Association, Kenyans in Alberta Association, Nigeria Association, and the Africa Centre. Participants were eligible if they were: (1) Recent immigrants (mostly within the past 10 years); (2) were of African descent and identify as Black; (3) had lived in Edmonton for not less than 2 years; (4) were aged 18 years or older; and (5) were proficiency in English language and gave informed consent.
Surveys: The survey study (n=119) instrument was designed to assess multiple levels of barriers and facilitators to engagement in collective community gardening. Additionally, the questionnaire assessed the general trends in the population regarding engagement in urban agriculture and the type of gardening. The questionnaire was administered through Survey Monkey combining face-to-face interviews and telephone conversations for broader accessibility of the immigrants. The survey data were then exported into SPSS version 29 for analysis. Descriptive statistics, frequencies, and proportions were calculated to summarize the demographic characteristics of the sample and key variables.
One-on-one interviews: To obtain nuanced narratives, we conducted n=10 in-depth interviews (IDI) with purposefully selected participants from the survey sample. Participants were selected to ensure adequate representation, considering factors such as age, gender, ethnicity, and year of arrival. The interviews were designed as semi-structured, open-ended conversations, creating a "listening and conversational space" (Creswell & Poth, 2018) where participants could freely articulate their experiences, views, and narratives. Interviews lasted approximately 35-60 minutes each and were conducted virtually on zoom to accommodate participant schedules and preferences.
Afrocentric Sharing Circles: Participants were engaged in facilitated online sharing circles (n=2). In these sessions, we designed the process to create a safe space and encourage participants to engage authentically as Black people and to share their stories in a non-judgmental and culturally affirming space. This approach created space for dialogue and allowed us to probe for deeper insight where necessary. Participants shared personal and collective experiences of facilitators and barriers to engagement and general experiences with collective community gardening in Edmonton, Alberta.
All IDI and Afrocentric sharing circles were digitally recorded, transcribed verbatim, and imported into NVivo version 15. Transcripts were then iteratively analyzed deductively (using the socio-ecological framework), and inductively to allow new themes and sub-themes to emerge and to develop a standard set of codes describing patterns observed in the data. The coding process followed Braun and Clarke’s (Braun & Clarke, 2006) six-step for thematic analysis: (1) Data familiarization; (2) Initial code generation; (3) Theme identification; (4) Theme review; (5) Theme definition and naming; and (6) Report production. The coding process was iterative, with line-by-line content analysis. We integrated qualitative narratives with survey data to strengthen the validity and depth of our findings. This triangulation approach enabled a comprehensive analysis of participant experiences while preserving the richness of individual accounts.
Results – More discussion of the nature of collective community garden groups is warranted. I got the impression throughout that the community gardens were predominated or set up for the African immigrant community until I got to lines 370-378. Given the population focus of the article, it’s not too much of a logical leap to make, even if it is a leap or assumption. The setup, history, and demographics of these gardens should be described much earlier in the manuscript.
We have extensively revised the results section and incorporated the suggestions. Thanks so much for these great comments!
Additionally, I have noted several concerns in specific areas:
Line 11 – too many commas in this sentence
Line 18 – comma after “barriers” should be a semicolon
Lines 21-26 – This is a long, yet incomplete sentence. Really, the explanations for each category of the socio-ecological model belong in the literature review (e.g lines 114-130) instead of the abstract.
Thanks you for this suggested edits on the abstract. We have revised the abstract and made it more succinct. The Abstract now reads as follows:
Abstract: Background: Community gardens are increasingly popular in Canadian cities, serving as transformative spaces where immigrants can develop self-reliant strategies for accessing culturally familiar and healthy nutritious foods. However, numerous facilitators and barriers exist that limit the engagement of racialized groups such as the Black identifying immigrants. Using a socio-ecological framework, this research explores barriers and facilitators of engagement of Black African immigrants in collective community gardening by in Alberta, Canada. Methods: The study adopted a community-based participatory research (CBPR) approach using mixed-methods. Data collection included structured surveys (n=119) to assess general engagement, facilitators, and barriers; in-depth interviews (n=10) to explore lived experiences, and Afrocentric sharing circles (n=2) to probe collective perspectives in relation to engagement in collective gardening. Participants were purposefully recruited through community networks within African immigrant-serving community organizations. Results: Our findings revealed how barriers at various levels of the socio-ecological model (SEM) interact to shape interest and engagement of African immigrants in collective community gardening. Access to collective gardens was associated with significant benefits, including maintaining healthy foodways, knowledge exchange, growing social capital, and community connections that support overall wellbeing. Conclusions: This study contributes an accessible framework for understanding and addressing the complex barriers that limit engagement in community gardens for vulnerable communities, while highlighting opportunities for creating more inclusive and culturally responsive urban agriculture initiatives.
Introduction section
Lines 62-70 – I’m confused by this paragraph. The authors state that there is a diference between collective and community gardens, yet go on to define “collective community gardens” as a concept leaving those terms undiferentiated. Throughout, “collective,” “community,” and “community collective” are used seemingly interchangeably. Likewise, your participants all seem to know what a “collective community garden” is, so it seems to be an emic term stemming either from the population of interest or the organization running the gardens. There needs to be more discussion of why the participants refer to it in this way.
Line 78 – “A similar study” has an unclear reference point. Similar to the current study, or to what’s referenced in the previous sentence?
Line 89 – Add comma after “gardens”
Lines 96-98 – Here, it is worth diferentiating what the diference between Black Canadians and recent African immigrants are. For example, that much of the Black population as designated by the Canadian census have Caribbean roots and more recent migrants come directly from Africa. Therefore, they relate to their new home in diferent waves, whether it be a diference in historically being subjected to slavery versus colonialism, or their cultural familiarity with Canadian customs. Since the study site is Edmonton, a description of the Black population in that city (or Alberta in general) is warranted. I see Kenya mentioned later on, but what other countries go into the African immigrant population?
Which are the most common countries of origin? This is not even touched on until the Limitations section.
Line 106 – no comma after “this study”
Thanks for these excellent suggestions on how we could improve the introduction section. We have incorporated all the suggestions and the introduction section reads as follows:
- Introduction
Community gardens are an increasingly popular model of urban agriculture in cities in the Global North and are serving as transformative and restorative spaces for individuals and populations but more specifically, for immigrants and refugees (Hartwig & Mason, 2016; Hume et al., 2022; Siegner et al., 2018). Different models of urban agriculture exists including the traditional community gardens, backyard gardening, and collective community gardens (Drake & Lawson, 2015; Guitart et al., 2012). For the purpose of this study, our focus is on collective community gardens. We use collective community gardens to refer to community gardens operated through shared ownership, collaborative labor, and communal harvest distribution. In these spaces, decisions about what to plant, how to maintain the garden, and how to distribute produce are made collectively by the community members (Guitart et al., 2012). This model and more generally, community gardens have numerous benefits.
Recent evidence in migration and food security literature show that engagement in collective community gardening creates opportunities for immigrants to reshape their relationships with the food system in host countries (Agustina & Beilin, 2012; Carney et al., 2012; Gerodetti & Foster, 2023; Hartwig & Mason, 2016; Hume et al., 2022; Onyango et al., 2025). These gardening spaces provide self-reliance strategies for accessing healthy food options and producing culturally familiar foods (Drake & Lawson, 2015; Gerodetti & Foster, 2023; Hite et al., 2017; Hume et al., 2022; C. Lindner, 2021; Minkoff-Zern et al., 2024). Additionally, evidence have revealed that engagement in collective community gardening contributes to improved health and successful integration of immigrant and refugee communities (Harris et al., 2014; Hume et al., 2022; Tracey et al., 2023). This trend has been observed across cities where urban gardens have served as transformative spaces that bridge cultural divides while promoting food security and embodied well-being (Lampert et al., 2021; Nisbet et al., 2022). Beyond nutritional benefits, these spaces cultivate neighborly connections and the recreation of social supports for newcomers.
However, collective community gardens face several barriers including bureaucracies in land access, fragmented regulations, infrastructure limitations, and varying and/or competing priorities within communities and in responsible government departments (Kwartnik-Pruc & Droj, 2023). For example, a recent study in the United States of America (USA) indicated that land access including the physical location of the community gardens are important predictors of interest and participation (Butterfield, 2022). Another study that focused on an exploration of experiences of South Asian refugee and immigrants found that barriers to participation in collective community gardening manifest in unique ways including spatial and transportation challenges (Hartwig & Mason, 2016). However, approaches that center the lived experiences of participants, especially immigrants, and how they perceive and use these spaces can inform more inclusive collective community gardens (Dyg et al., 2020; Nisbet et al., 2022).
Recent studies emerging from the Canadian context have recommended accessible green spaces, including collective community gardens, as important resources for mental health and wellbeing (Charles-Rodriguez et al., 2023; Hordyk et al., 2015; C Lindner, 2021). Unfamiliarity with the local environment and the cold Northern climate have been identified as barriers to engagement in community gardening in Canadian cities; yet, opportunities for obtaining healthy food, building new relationships, and emotional restoration remain key motivations that outweigh those barriers to participation (Harris et al., 2014; Hordyk et al., 2015). Despite the known benefits of collective community gardening, especially for minoritized communities, immigrants in Canada continue to experience an array of barriers to engagement in urban agriculture (Jefferies et al., 2022; Onyango et al., 2025). For example, newcomers to Canada may have advanced farming knowledge in the context of the countries of origin but limited knowledge on how the appropriate garden in Canada. Additionally, recent reviews of existing literature have revealed these individuals tend to experience a downward socioeconomic mobility, requiring them to work multiple minimum wage jobs to make ends meet coupled with other challenges, hence limited time to engage in collective community gardening (Jefferies et al., 2022; Mori & Onyango, 2023; PROOF, 2024). Based on our knowledge, limited to no primary research exists on the barriers to engagement in collective community gardening, particularly, for the Black identifying immigrants who have reported an increased risk to food insecurity (Jefferies et al., 2022; Mori & Onyango, 2023; PROOF, 2024).
Our study aimed to systematically address this knowledge gap by exploring the facilitators and barriers to engagement in collective community gardening. The focus of the study was on how the gardening spaces can better serve to maintain healthy foodways post-migration and cultivate social connections that support wellbeing and integration of African immigrants in Alberta, Canada. This study was guided by the research question: What are the barriers and facilitators to engagement in collective community gardening among Black African immigrants settled in Alberta, Canada? The study was guided by the socio-ecological model to explore the individual, interpersonal, community, environmental, and the systemic barriers and facilitators of engagement of immigrants.
Theoretical framework - Socio-ecological model
The socio-ecological framework (McLeroy et al., 1988; Stokols, 1996) informed the conceptualization of this research examining barriers and facilitators to engagement in collective community gardening. The model incorporates the interactive effects of individual, interpersonal, community, environmental, and structural factors to examine participation in gardening and to help understand the intersecting barriers operating at different spheres of influence (Drake & Lawson, 2015; Hartwig & Mason, 2016). The framework is particularly relevant for immigrant communities as it will uncover how the personal factors (like time and resource limitations) interact with interpersonal dynamics (such as cultural connections and discrimination) and broader structural constraints (including systemic inequities) to influence participation in collective community gardening (Hartwig & Mason, 2016). At the individual level, we examine micro-level factors such as demographic characteristics, time constraints, gardening knowledge and skills, and economic circumstances. The interpersonal level encompasses social networks, cultural connections, experiences of discrimination, and relationship dynamics within garden spaces. Community-level factors include garden accessibility, decision-making processes, and organizational characteristics of collective community gardens. Environmental factors consider climate constraints, growing seasons, and physical infrastructure, while structural factors examine systemic barriers like immigration policies, the economic environment, and resource allocation.
Lines 148-151 – What did the Sinkunia CDO get out of this project? In other words, why did they agree to play such a key role in this study? Non-profits with limited resources don’t tend to expend them on helping researchers unless 1) the researchers are contributing to programming through volunteering or 2) they anticipate researcher products which can help their operations.
Line 181 – How were participants selected for interviews? “Purposefully” is too vague. Did you, for example, select for newer and established gardeners? A variety of diferent national origins?
We have extensively revised the methods section and incorporated the suggestion! Thanks so much for these valuable suggestions.
Lines 188-194 – Readers need some more information on what Afrocentric circles look like. I hesitate to say “cite the literature” because that feels like imposing a very Western-centric means of defining something through a formal capture into written tradition. Regardless, not being familiar with this method I was unable to come away with a clear picture of what it entailed, so you need a richer description of it.
We have revised the details on Afrocentric Sharing Circle to highlight what it looks like and hwo the sharing circles were facilitated. A section of the methods now reads as follows:
Afrocentric Sharing Circles: Participants were engaged in facilitated online sharing circles (n=2). In these sessions, we designed the process to create a safe space and encourage participants to engage authentically as Black people and to share their stories in a non-judgmental and culturally affirming space. This approach created space for dialogue and allowed us to probe for deeper insight where necessary. Participants shared personal and collective experiences of facilitators and barriers to engagement and general experiences with collective community gardening in Edmonton, Alberta.
Line 234 – Should be “participated” to keep a consistent tense.
Thanks for this comment! Have revised the tense and reviewed the entire paper for any inconsistencies in tenses to the best of our knowledge. I hope it reads much better now!
Line 238 – male and female centered should either both have or not have a dash connecting them. Additionally, this needs more elaboration. Were participants asked to self-report the structure of their household? Some explanation of how this was presented to them is warranted.
Well noted and have adjusted accordingly! The results in Table 1 are from the survey results and participants were asked about their household structure this was checked appropriately based on their description/response.
Line 249 – Figure 1 requires a listing of N values. This is because in surveys not everybody responds to every question, so typically you want an N for each question.
Line 424 – Sentence should start with “A short growing season…” Line 477 – No apostrophe needed on “gardens”
Thanks for pointing this out. We have revised this section in the results section.
Line 478 – Semicolon before “however”
Lines 492-495 – Discussion on gendered relationships to community gardening is better suited for the results section, with supporting data. The discussion of gender should be more expansive or saved for another paper. As it stands in the current manuscript, it reads like an afterthought and that does the significance of it a disservice.
Lines 527-529 – This is an incomplete sentence.
Line 540 – Is “cultural anchors” in reference to another publication, or is it meant to be a term applied to the current work? This is highly confusing because either way it seems like the authors are introducing major idea without due service to it. It should either be an organizing concept in the paper or omitted.
Thanks you for all these great suggestions on how to improve the discussion section. We have revised the discussion section, which now reads as follows:
- Discussion
This study has generated novel insights into the complex interplay of the factors influencing Black African immigrants' engagement in collective community gardening. The facilitators identified in this study illustrate the multi-functional benefits of collective community gardening for immigrant communities. A strong connection to agriculture and food production, often rooted in pre-migration experiences, is a primary facilitator. Most immigrants have prior connections to agriculture and more specifically, to the land, which then acts as a motivation for their engagement in gardening (Brooks & Marten, 2012; Butterfield, 2022; Charles-Rodriguez et al., 2023; Onyango et al., 2025). These findings are consistent with previous studies that have demonstrated that most immigrants are motivated by the desire to maintain their traditional foodways and to reconstruct their social connection through engagement on the garden (Elshahat et al., 2024; Ghose & Pettygrove, 2014; Minkoff-Zern et al., 2024).
Additionally, the perceived health benefits including access to fresh produce, physical activity, and improved mental wellbeing align with previous research on the role of community gardens in embodied wellbeing and social integration of immigrants (Hume et al., 2022; Siegner et al., 2018). Engagement in collective community gardening created opportunities for individuals in this study to experiment with traditional crop and vegetable varieties, maintain healthy food cultures, and building social capital through societal connections. Such social networks also support newcomer resettlement through meeting new people, learning about Canadian culture, and developing context specific agricultural skills. These findings suggest that local governments and immigrant-serving organization should incorporate culturally syntonic initiatives such as collective community gardens into their programming; policymakers have the responsibility to expand growing spaces in immigrant-dense residential areas.
Evidence from the current study also show that collective community gardens are helpful community resources. Participants who were actively engaged in collective gardening often did so through existing community connections. Meeting people who share similar experiences and backgrounds emerged as a key motivation. Our analysis reveals two roles collective community gardens play: building social capital and knowledge exchange. Social capital facilitates initial access and supports knowledge transfer about growing food in Canada. These findings extend previous work by Kingsley et al. (Kingsley et al., 2020) on the role of social networks in immigrant food systems by demonstrating how these relationships play out in spaces of urban agriculture.
However, limited access to land, irrigation facilities, and other essential inputs may constrain people’s interest to engage. At the individual level, time constraints emerge as a significant challenge, reflecting the broader challenges of immigrant families balancing multiple jobs and responsibilities. Most immigrants have multiple pressures competing for limited time and resources. The emphasis on financial pressures and childcare responsibilities reveals how individual barriers are often manifestations of broader structural inequity. Economic pressures create systemic barriers influencing such individuals' ability to engage in community-driven initiatives. Most participants lacking information about collective community gardening opportunities suggests current efforts may not effectively reach some immigrant communities, particularly recent arrivals. Language barriers reported by some participants is a significant barrier to resource navigation and accessibility.
Individual barriers also intersect with interpersonal dynamics in collective community gardens, where experiences of discrimination and cultural exclusion can create additional barriers to engagement in collective community gardening. This finding adds nuance to existing literature on community gardens as sites of social integration (Agustina & Beilin, 2012; Carney et al., 2012; Gerodetti & Foster, 2023; Ghose & Pettygrove, 2014; Hartwig & Mason, 2016; Hume et al., 2022), suggesting that without intentional attention to cultural sensitivity and inclusion, these spaces may inadvertently reproduce broader social and structural patterns of inequity. Furthermore, these findings highlight the need for flexible collective community garden programming that accommodates diverse work schedules and family commitments of immigrant participants. Immigrant-serving organizations should consider outreach strategies through trusted community networks to address information gaps, in addition to offering multilingual resources and childcare support during garden activities. Future research should examine how successful garden initiatives navigate these multilayered barriers and identify effective strategies for creating more accessible and welcoming collective garden spaces for diverse immigrant communities.
Analysis of community-level factors reveals how representation, accessibility, and inclusive decision-making concerns shape interest and engagement in collective community gardening. The organizational structure of collective gardens significantly influences participation. Non-participatory decision-making can create barriers to meaningful engagement. Such models may exclude certain perspectives, whereas inclusive decision-making processes that involve participants in determining what and how crops are grown can create welcoming spaces that reflect the diversity of experiences, traditions, and perspectives that enrich collective community gardens. These findings suggest the need for regular opportunities for the gardening participants to provide feedback on structure and activities, with transparent processes for implementing community suggestions.
In addition, findings from this study raises important questions about the viability of collective community gardening as a strategy for addressing cultural food insecurity in Northern climates of Canadian Prairie Provinces such as Alberta. Edmonton's short growing season (May to mid-September) creates significant challenges for cultivating tropical vegetables that typically require longer. These environmental barriers are compounded by limited access to protected growing spaces like greenhouses, making year-round production of culturally familiar vegetables nearly unfeasible. Climate change further exacerbates these challenges, with increasing weather unpredictability particularly affecting those without access to protected growing spaces (Leung, 2023). However, investment in infrastructure such as community greenhouses could transform these spaces from seasonal projects into sustainable, year-round resources for food production, community building, and neighborhood revitalization.
Our findings reveal how systemic racism operates as a structural barrier affecting multiple aspects of participation in the gardening activities. This manifests in both subtle and overt ways, from the distribution of garden infrastructure to decision-making processes about what should be grown. Additionally, economic structures create pervasive barriers to participation on the garden. Despite high educational attainment among immigrants, many face significant employment challenges that directly affect their ability to engage in community gardening. In the case of community gardening, systemic racism could also influence resource allocation, land use priorities, program design, and distribution of gardening infrastructure. Community gardens in Edmonton, like most cities in North America, show a pattern of concentration in predominantly non-immigrant neighborhoods, reflecting and reinforcing existing spatial inequities (Jefferies et al., 2022; Li et al., 2023; Minkoff-Zern et al., 2024; Nisbet et al., 2022). Furthermore, funding models and program requirements may downplay the needs and circumstances of immigrant communities, therefore, creating additional barriers to meaningful engagement.
While community gardens are resources for community and neighborhood revitalization, the current structural environment creates pervasive barriers that exclude the communities that could benefit the most from collective community gardens. While improved information sharing and culturally appropriate programming are important, meaningful change requires addressing the underlying systemic inequities that create barriers to participation. This should include policy changes around seed importation, investment in greenhouse infrastructure in immigrant neighborhoods, reform of land use regulations, and strengthening ongoing efforts to address economic and racial inequity.

Reviewer 2 Report
Comments and Suggestions for Authors
Please see my comments in the attached file.

Please see my comments in the attached file.
Author Response
Reviewer #1
Title: Barriers and Facilitators to Engagement in Collective Gardening among Black African Immigrants in Alberta, Canada
The paper is overall written well and presents interesting insights from a Community-Based Participatory study with Black African Immigrants in Alberta, Canada in relation to collective community gardening. More detailed comments are offered below for enhance clarity in order to reach publication stage.
G eneral:
- There are some typographical issues which needs correction throughout the
Thanks for this comment! We have reviewed and edited the paper to clean up any editorial/typological errors!
- Please pay close attention to repetition of points across various section to reduce redundancy to shorten the A very long paper can overwhelm a reader.
Thanks for pointing this out! We have reviewed and deleted areas of repetition that we identified!
- The application of socioecological model (SEM) should be similar for both barriers and In the current version, the Result section has applied it to barriers only. This is inconsistent with description in Discussion section about the use of SEM – e.g., authors state “Applying a socio- ecological model (SEM), we identified multiple interconnected barriers and facilitators.” (line 459- 460).
A bstract:
- Background: please make it concise
- Results: the sentence starting with “Our findings demonstrate how various levels of the socioecological model (SEM) – individual…” is
Thank you for this great feedback! We have revised the abstract and now reads as follows:
Abstract: Background: Community gardens are increasingly popular in Canadian cities, serving as transformative spaces where immigrants can develop self-reliant strategies for accessing culturally familiar and healthy nutritious foods. However, numerous facilitators and barriers exist that limit the engagement of racialized groups such as the Black identifying immigrants. Using a socio-ecological framework, this research explores barriers and facilitators of engagement of Black African immigrants in collective community gardening by in Alberta, Canada. Methods: The study adopted a community-based participatory research (CBPR) approach using mixed-methods. Data collection included structured surveys (n=119) to assess general engagement, facilitators, and barriers; in-depth interviews (n=10) to explore lived experiences, and Afrocentric sharing circles (n=2) to probe collective perspectives in relation to engagement in collective gardening. Participants were purposefully recruited through community networks within African immigrant-serving community organizations. Results: Our findings revealed how barriers at various levels of the socio-ecological model (SEM) interact to shape interest and engagement of African immigrants in collective community gardening. Access to collective gardens was associated with significant benefits, including maintaining healthy foodways, knowledge exchange, growing social capital, and community connections that support overall wellbeing. Conclusions: This study contributes an accessible framework for understanding and addressing the complex barriers that limit engagement in community gardens for vulnerable communities, while highlighting opportunities for creating more inclusive and culturally responsive urban agriculture initiatives.
IIIIntroduction:
- Para 2 and Para 3 can be merged to reduce
- The authors introduce the term “community garden” and “collective community garden” in Para 4 (line 62). However, in Para 5 the authors go back to talk about “community garden” in general; this makes a reader wonder why to distinguish the two terms in Para
- Could you please review closely the use of term “community garden” in line 89? Are you referring to collective community garden here?
- Line 95: use “vulnerable population” instead of “high-risk population”
Thank you for this great feedback! We have incorporated the suggestions throughout the introduction section, which now reads as follows:
- Introduction
Community gardens are an increasingly popular model of urban agriculture in cities in the Global North and are serving as transformative and restorative spaces for individuals and populations but more specifically, for immigrants and refugees (Hartwig & Mason, 2016; Hume et al., 2022; Siegner et al., 2018). Different models of urban agriculture exists including the traditional community gardens, backyard gardening, and collective community gardens (Drake & Lawson, 2015; Guitart et al., 2012). For the purpose of this study, our focus is on collective community gardens. We use collective community gardens to refer to community gardens operated through shared ownership, collaborative labor, and communal harvest distribution. In these spaces, decisions about what to plant, how to maintain the garden, and how to distribute produce are made collectively by the community members (Guitart et al., 2012). This model and more generally, community gardens have numerous benefits.
Recent evidence in migration and food security literature show that engagement in collective community gardening creates opportunities for immigrants to reshape their relationships with the food system in host countries (Agustina & Beilin, 2012; Carney et al., 2012; Gerodetti & Foster, 2023; Hartwig & Mason, 2016; Hume et al., 2022; Onyango et al., 2025). These gardening spaces provide self-reliance strategies for accessing healthy food options and producing culturally familiar foods (Drake & Lawson, 2015; Gerodetti & Foster, 2023; Hite et al., 2017; Hume et al., 2022; C. Lindner, 2021; Minkoff-Zern et al., 2024). Additionally, evidence have revealed that engagement in collective community gardening contributes to improved health and successful integration of immigrant and refugee communities (Harris et al., 2014; Hume et al., 2022; Tracey et al., 2023). This trend has been observed across cities where urban gardens have served as transformative spaces that bridge cultural divides while promoting food security and embodied well-being (Lampert et al., 2021; Nisbet et al., 2022). Beyond nutritional benefits, these spaces cultivate neighborly connections and the recreation of social supports for newcomers.
However, collective community gardens face several barriers including bureaucracies in land access, fragmented regulations, infrastructure limitations, and varying and/or competing priorities within communities and in responsible government departments (Kwartnik-Pruc & Droj, 2023). For example, a recent study in the United States of America (USA) indicated that land access including the physical location of the community gardens are important predictors of interest and participation (Butterfield, 2022). Another study that focused on an exploration of experiences of South Asian refugee and immigrants found that barriers to participation in collective community gardening manifest in unique ways including spatial and transportation challenges (Hartwig & Mason, 2016). However, approaches that center the lived experiences of participants, especially immigrants, and how they perceive and use these spaces can inform more inclusive collective community gardens (Dyg et al., 2020; Nisbet et al., 2022).
Recent studies emerging from the Canadian context have recommended accessible green spaces, including collective community gardens, as important resources for mental health and wellbeing (Charles-Rodriguez et al., 2023; Hordyk et al., 2015; C Lindner, 2021). Unfamiliarity with the local environment and the cold Northern climate have been identified as barriers to engagement in community gardening in Canadian cities; yet, opportunities for obtaining healthy food, building new relationships, and emotional restoration remain key motivations that outweigh those barriers to participation (Harris et al., 2014; Hordyk et al., 2015). Despite the known benefits of collective community gardening, especially for minoritized communities, immigrants in Canada continue to experience an array of barriers to engagement in urban agriculture (Jefferies et al., 2022; Onyango et al., 2025). For example, newcomers to Canada may have advanced farming knowledge in the context of the countries of origin but limited knowledge on how the appropriate garden in Canada. Additionally, recent reviews of existing literature have revealed these individuals tend to experience a downward socioeconomic mobility, requiring them to work multiple minimum wage jobs to make ends meet coupled with other challenges, hence limited time to engage in collective community gardening (Jefferies et al., 2022; Mori & Onyango, 2023; PROOF, 2024). Based on our knowledge, limited to no primary research exists on the barriers to engagement in collective community gardening, particularly, for the Black identifying immigrants who have reported an increased risk to food insecurity (Jefferies et al., 2022; Mori & Onyango, 2023; PROOF, 2024).
Our study aimed to systematically address this knowledge gap by exploring the facilitators and barriers to engagement in collective community gardening. The focus of the study was on how the gardening spaces can better serve to maintain healthy foodways post-migration and cultivate social connections that support wellbeing and integration of African immigrants in Alberta, Canada. This study was guided by the research question: What are the barriers and facilitators to engagement in collective community gardening among Black African immigrants settled in Alberta, Canada? The study was guided by the socio-ecological model to explore the individual, interpersonal, community, environmental, and the systemic barriers and facilitators of engagement of immigrants.
Theoretical framework - Socio-ecological model
The socio-ecological framework (McLeroy et al., 1988; Stokols, 1996) informed the conceptualization of this research examining barriers and facilitators to engagement in collective community gardening. The model incorporates the interactive effects of individual, interpersonal, community, environmental, and structural factors to examine participation in gardening and to help understand the intersecting barriers operating at different spheres of influence (Drake & Lawson, 2015; Hartwig & Mason, 2016). The framework is particularly relevant for immigrant communities as it will uncover how the personal factors (like time and resource limitations) interact with interpersonal dynamics (such as cultural connections and discrimination) and broader structural constraints (including systemic inequities) to influence participation in collective community gardening (Hartwig & Mason, 2016). At the individual level, we examine micro-level factors such as demographic characteristics, time constraints, gardening knowledge and skills, and economic circumstances. The interpersonal level encompasses social networks, cultural connections, experiences of discrimination, and relationship dynamics within garden spaces. Community-level factors include garden accessibility, decision-making processes, and organizational characteristics of collective community gardens. Environmental factors consider climate constraints, growing seasons, and physical infrastructure, while structural factors examine systemic barriers like immigration policies, the economic environment, and resource allocation.
T
M aterials and Methods
- Line 146 is Authors write “We employed a community-based participatory research (CBPR) approach guided 146 by principles outlined by [46] and [47].” Please remove “by” and “and” before citation [46] and [47].
- After line 160, introduce your mix method approach to create a transition to next “Participant Recruitment” section, which has details of each of three methods you used. For example, you can add a sentence after line 160: “A mixed methods approach was employed by using online survey, in- depth interviews and sharing circles”.
We have revised the Research Design section, which now reads:
Research Design
We employed a community-based participatory research (CBPR) approach guided by principles of equitable partnership, co-learning, and community benefit (Israel et al., 1998; Minkler & Wallerstein, 2008). The study was conducted in Edmonton, Alberta, a popular destination for immigrants of African descent in Canada (Statistics Canada, 2021). Our community partner, Sinkunia Community Development Organization (SCDO), an African-led non-profit organization, played a crucial role in shaping research objectives and facilitating community engagement. Sinkunia CDO operates various settlement programs and provides culturally appropriate support services for African immigrants in Alberta, including initiatives like a collective community garden and partnership with the Edmonton Food Bank to improve access to culturally appropriate supports. A mixed methods approach was employed by using online survey, in- depth interviews and sharing circles. In this CBPR project, we maintained continuous engagement through regular meetings and feedback sessions with our community partner who was involved in the design, implementation, analysis, and knowledge mobilization (Lincoln & Guba, 1985; Minkler & Wallerstein, 2008).
- Participant Recruitment: please correct grammatical errors when describing eligibility criteria; sometimes verb ‘were’ is used and other times ‘are’ is Similar error is found at some other place (e.g. line 171.)
Great suggestion! We have revised Participant recruitment section and now reads as follows:
Participant Recruitment
Participants were purposefully recruited in collaboration with SCDO and existing community networks, including the Somali Cultural Association, Kenyans in Alberta Association, Nigeria Association, and the Africa Centre. Participants were eligible if they were: (1) Recent immigrants (mostly within the past 10 years); (2) were of African descent and identify as Black; (3) had lived in Edmonton for not less than 2 years; (4) were aged 18 years or older; and (5) were proficiency in English language and gave informed consent.
- Did participants receive any honorarium?
Thanks for this valuable comments. This has been incorporated in the ethics section which now reads as follows:
Ethics and Participant Protection:
The study received approval from the University of Alberta Research Ethics Review Board (Protocol ID: Pro00134979). After receiving information about study objectives, benefits, and potential risks, participants provided informed consent (written or verbal). With permission, virtual interviews were recorded. To ensure confidentiality, we collected no personal identifiers, and participants had complete autonomy to decide on the details of information they wished to share. Participation was voluntary, and participants had the right to withdraw at any time. Participants each received honoraria of CAD$25 for survey completion and CAD$35 for in-depth interviews or Afrocentric sharing circles to acknowledge their time and contributions to the study. All data was de-identified and securely stored following the University of Alberta data management guidelines.
- Were in-depth interviews in-person or virtual? Please make it clear for line 180-181.
The IDI section has been revised and now reads as follows:
One-on-one interviews: To obtain nuanced narratives, we conducted n=10 in-depth interviews (IDI) with purposefully selected participants from the survey sample. Participants were selected to ensure adequate representation, considering factors such as age, gender, ethnicity, and year of arrival. The interviews were designed as semi-structured, open-ended conversations, creating a "listening and conversational space" (Creswell & Poth, 2018) where participants could freely articulate their experiences, views, and narratives. Interviews lasted approximately 35-60 minutes each and were conducted virtually on zoom to accommodate participant schedules and preferences.
- “Ethics and Participant Protection” Please rephrase “…and participants controlled their level of disclosure”. Do you mean “participants had complete autonomy to decide on the details of information they wished to share”?
The ethics section has been revised and now reads as follows:
Ethics and Participant Protection:
The study received approval from the University of Alberta Research Ethics Review Board (Protocol ID: Pro00134979). After receiving information about study objectives, benefits, and potential risks, participants provided informed consent (written or verbal). With permission, virtual interviews were recorded. To ensure confidentiality, we collected no personal identifiers, and participants had complete autonomy to decide on the details of information they wished to share. Participation was voluntary, and participants had the right to withdraw at any time. Participants each received honoraria of CAD$25 for survey completion and CAD$35 for in-depth interviews or Afrocentric sharing circles to acknowledge their time and contributions to the study. All data was de-identified and securely stored following the University of Alberta data management guidelines.
RResults
- Line 223:-225. “As detailed in Table 1, the majority (55%) of respondents were female, this trend was also observed in general garden participation as more females were engaged in collective community gardening than ” Is there a citation for the trend you are referring to? The source of information you are referring here is unclear.
Thanks you for raising this! We have edited the results section and now reads as follows:
- Results
In this section, we present the demographic information of the study participants and the common types of community gardens reported by the participants. Subsequently, this is followed by a synthesis of the facilitators and barriers to engagement in collective community gardening
3.1. Socio-demographic characteristics of survey participants
Table 1, presents the demographic characteristics of survey study participants. More than half of the participants (55%), were female most of who engage in backyard/home gardening (41.2%) and collective community gardening (37.0%). On average, the participants were relatively young and had a median age of 39 years, indicating additional responsibilities with significant time and resources dedicated to childcare. Furthermore, most participants had migrated within the last 10 years with at least 7 in 10 participants having post-secondary educated, spanning some college to a post-graduate degree. However, only about half were formally employed. One in five had more than one job, indicating that such highly educated newcomer individuals, still navigating socioeconomic and cultural changes and are taking up multiple low wage jobs to make ends meet.
Table 1. Participant Demographics of Online Survey
|
Survey participant demographics, n=119 |
|
|
Gender, n (%) Female Male |
65 (54.6) 54 (45.4) |
- There are errors here with use of verb ‘were/are’; please edit
We have reviewed and edited the areas with tense variation through out the paper!
- Line You have used term collective community garden two times and the % does not match with
% in Table 1. You state “collective community gardens (18.7%), and collective community (37.0%).”
Thanks for highlighting this error! We have revised the section and deleted the repeated words.
- Table For the variable “Common types of gardening, (%0”, please add a footnote that respondents were able to select more than one response. Currently, the total for % of each response sum up to 124%
Well noted! Thank you for pointing this out! We have revised the table and a section reads as follows:
|
Household Structure, n (%) Female centered Male centered Nuclear Extended |
28 (23.5) 21 (17.6) 54 (45.4) 16 (13.4) |
|
|
Common types of gardening, (%)* Home/backyard gardening Collective community gardens Community gardens No identified |
49 (41.2) 44 (37.0) 33 (27.7) 22 (18.5) |
|
|
*Participants were allowed to select more than one response option |
|
|
- Section “3.2. Facilitators of engagement in collective community gardening”. Why was socioecological model (SEM) not applied here? This is a big missing piece and causing inconsistency in analysis of facilitators” and
Thank you for pointing this out! We have incorporated this comment in our analysis and presentation of the facilitators and barriers to engagement of immigrants in collective community gardening. The section on facilitators now reads as follows:
3.2. Facilitators of engagement in collective community gardening
Survey findings showed that most immigrants engage in collective community gardening to gain community connections and to build new relationships. Other notable facilitators included having access to land, knowledge, and resources, as well as being supported by community organizations. (See Figure 1 below).
Figure 1. Proportion of participants reporting facilitators to engagement in collective community gardening
Analysis and synthesis of qualitative data revealed the individual, interpersonal, community, and environmental level facilitators of engagement. At the individual level, immigrants highlighted a) connections to agriculture and food production; and b) the associated health and wellbeing benefits as potential motivators. At the interpersonal and community level facilitators, c) the social connections and opportunity to meet new people coupled with the existence of the necessary resources such as land and established community gardens in neighborhoods.
Connection to agriculture and cultural foods: It was observed that for some participants, a perceived connection to growing food was a key motivation. This was notable among participants who expressed how agriculture was a part of their identity in their home countries before migration. Therefore, engaging in collective community gardening provided a sense of continuity, and place-making through maintenance of traditional foodways post-migration. Participants expressed strong interest in growing culturally familiar vegetables and herbs, viewing collective community gardens as potential spaces for cultural food sovereignty.
I have been involved in collective community gardening. I love gardening a lot, unfortunately, I didn’t know about it earlier. It is one of the best things one can ever do. Any community or any culture that treasures farming, they can thrive in these spaces. They don’t need to depend on external help for survival. But if people become lazy, and they think that everything should come from their government and sit down and wait for the government to do something, then it’s a recipe for disaster, for the people, or for the community. So yes, I have been involved, and I would love to continue. (IDI-06)
I can say it’s important to me, partly because these are the foods that I have grown up with, and also because I understand the nutritional value of some of these foods, especially the vegetables. (IDI-02)
Perceived health benefits of engagement in collective community gardening: Participants expressed multiple perceived benefits and motivation of engaging in collective community gardening, including access to fresh organic vegetables, physical activity, and social supports. Gardening spaces create opportunities for people to have a sense of belonging by being part of a community activity that not only enhances the health and wellbeing of individuals but also that of the community at large.
Collective community gardening comes with a lot of benefits. To start with, you meet new people, people from different backgrounds and you form a community. We have different communities that we belong to. But you become part of a community, another community that is different from the ones that you’re familiar with. (IDI-05)
Meeting people, forming connections, and building social capital in the community: Underscored by participants who perceived collective community gardens as spaces where they can come out to meet people, cultivate new relationships, and nurture support for overall wellbeing and intercultural learning. Social capital played a role in engagement, with participants who knew someone already involved in collective community gardening expressing greater interest.
Because I once saw, my friend bring tomatoes from a collective community garden, not the one from the shop. From a real garden. It made me feel like, “Wow, is this being done here?” I was wondering how people were able to plant these things. So that got me curious that if there is a way, then maybe I will try it and learn from it. (IDI-04)
Building relationships within the community emerged as another motivation, collective community gardens are seen as welcoming spaces where knowledge exchange between and amongst community members and across generations.
In the collective community garden, because we have people from different countries, they come, and they’re like, “What are you growing there?” Vegetables that they’ve not seen, or they are like, “How do you prepare this?” Like when they see me grow a lot of kale, then they’re like, “What are you going to do with all that?”, then with that, we now start talking about, you know, how I use it, how I’ll prepare and store it. And yeah, so the collective community garden, it’s helped me learn quite a lot, meet people, and see how they are planting, you know, different ways of doing that because I learn something new. (IDI-02)
When gardens were accessible and culturally welcoming, they served as important and valued community spaces. Opportunities to meet other people who shared the same experiences in the collective community garden was identified as a key facilitator, cultivating such community and neighborly connections created an enabling environment where individuals felt welcome, appreciated, and valued.
- Section “3.3. Barriers to engagement in collective community gardening”. This section is very long. Please carefully re-read participant quotes and shorten them and/or remove some whenever possible. For example, line 389-393 where participant says “I had to find a way to create beds and grow in my backyard…” is not about collective community garden and could be removed. Likewise, line 408-411 starting with “...somebody might be like, ‘We are going to grow 20 heads of lettuce…”, is a made up/imaginary scenario that participant IDI-20 is This could be removed given that you have much stronger lived experience in the next quote by IDI-03. I appreciate the authors wish to share words of many participants and in detail, but the length of paper has become so long that it could compromise engagement of readers. I advice for a balanced approach please.
Well note and we appreciate your feedback. We have revised these sections of our results and the section now reads as follows:
Community-Level and Organizational Factors
Non-accessibility emerged as another barrier, over a third of survey respondents faced distance and transportation challenges.
I know some people who are not able to come to the collective community gardens because they are far from where they live. Like where we farm, for instance, I think it’s maybe more than 20 kilometers away from where we live, so it depends on, do people have time to go to the community garden, and how close is it? (IDI-02)
You know, people can even have spaces that are closer to where they live. People can decide that, “Oh, I would belong to the one that is closer to my house,” So it would be more convenient. Besides creating awareness, convenience is also another thing. My husband, for instance, has never come to the collective community garden. Because we live in the Southwest, and the collective community garden is on the Northside. My husband does not like going far. And even when we offer to take him, he’s like, “No, I’m not going.” So some people would not want to maybe drive ten minutes, 15 minutes, you know, between ten and 20 minutes to get to the collective community garden. So convenience is a barrier for some people. (IDI-05)
Non-participatory decision-making regarding what should be grown was identified as another barrier:
There is a time I had wanted to join a certain collective community garden. But then I was like, “When it comes to the collective, how do people make decisions on the type of vegetables or fruits to grow?” “How do you decide, or how do you agree that we are going to grow this type of vegetable and not this other one?”. So collective community gardens, it really depends on who is involved, because we have different tastes, different desires. In a collective garden, how do people make decisions on what to grow? (IDI-02)
I am involved with a collective community garden, here in Edmonton, but you find that all we do is to garden Canadian food, or even other foreign foods, and not African, but European foods. Because I remember we planted lots of garlic. They bring different species of garlic from Germany, Bulgaria, although they will say, “This one is German, this Bulgarian’s flavour,” all those species of garlic. And others, maybe like tomatoes, different types of tomatoes, they have the Italian, they have the Mexican, and things like that, but nothing from Africa. (IDI-03)
In addition, participants identified environmental constraints to collective community gardening in Edmonton’s Northern climate.
DDiscussion
- The discussion could be made short by removing some overlap with the Introduction and Results section. My concern here is again about keeping readers engaged until the end when a paper is very long.
Thanks for this valuable suggestion! We have made the discussion more succinct and reduced the redundancy in the write.
- At the end of each para, please add implications of the gained For example, how your findings could inform practice, policy and/or research.
Thanks for this valuable suggestion! We have added recommendations based on the evidence in each paragraph.
- Line 478-481. The sentence starting with “general interest…” is unclear or incomplete; Perhaps add word “helpful” to say “…collective community gardens are perceived as helpful community resources”.
This sentence has been revised and the section of the discussion now reads as follows:
- Discussion
This study has generated novel insights into the complex interplay of the factors influencing Black African immigrants' engagement in collective community gardening. The facilitators identified in this study illustrate the multi-functional benefits of collective community gardening for immigrant communities. A strong connection to agriculture and food production, often rooted in pre-migration experiences, is a primary facilitator. Most immigrants have prior connections to agriculture and more specifically, to the land, which then acts as a motivation for their engagement in gardening (Brooks & Marten, 2012; Butterfield, 2022; Charles-Rodriguez et al., 2023; Onyango et al., 2025). These findings are consistent with previous studies that have demonstrated that most immigrants are motivated by the desire to maintain their traditional foodways and to reconstruct their social connection through engagement on the garden (Elshahat et al., 2024; Ghose & Pettygrove, 2014; Minkoff-Zern et al., 2024).
Additionally, the perceived health benefits including access to fresh produce, physical activity, and improved mental wellbeing align with previous research on the role of community gardens in embodied wellbeing and social integration of immigrants (Hume et al., 2022; Siegner et al., 2018). Engagement in collective community gardening created opportunities for individuals in this study to experiment with traditional crop and vegetable varieties, maintain healthy food cultures, and building social capital through societal connections. Such social networks also support newcomer resettlement through meeting new people, learning about Canadian culture, and developing context specific agricultural skills. These findings suggest that local governments and immigrant-serving organization should incorporate culturally syntonic initiatives such as collective community gardens into their programming; policymakers have the responsibility to expand growing spaces in immigrant-dense residential areas.
- Conclusion This should be just one paragraph. Currently, some of the results and discussion is repeated here. You also bring some implications, but these should be part of the main Discussion section (after each para on gained insights).
Thanks! We have revised the conclusion section and it now reads as follows:
- Conclusions
In conclusion, this study makes significant contributions to literature and practice. The study highlights the relevance of collective community gardening in addressing social isolation, cultural food insecurity, and supporting mental health and wellbeing. However, a number of facilitators and barriers influence the participation of immigrants in collective community gardens. The facilitators and the barriers operate at different levels – individual, interpersonal, community, environmental, and structural spheres. Cutting across the various levels are factors such as – connection to agriculture and food production, the associated health and wellbeing benefits such as access to fresh produce, and the opportunities to build new connections as some possible motivations for immigrants. With reference to the barriers, busy schedules emerged as a common individual barrier, which reflects broader structural challenges around employment precarity and economic pressures facing immigrant communities. Our findings highlight how social capital operates as both barrier and facilitator, with community connections simultaneously enabling garden access for some while reinforcing exclusion for others. Extreme climate conditions, coupled with increasingly unpredictable weather are some critical barriers to engagement in urban gardening. Investment in community greenhouse infrastructure could extend the short growing seasons, protecting vegetables from increasingly volatile weather, and enabling year-round participation. Such infrastructure could transform collective community gardens from seasonal projects into sustainable, year-round spaces for food production, community building, and neighborhood revitalization. As Canadian cities become increasingly diverse, equitable access to collective community gardens is crucial for supporting immigrant well-being and social integration. This study contributes an accessible framework for understanding and addressing the complex barriers that can limit access for vulnerable communities, while highlighting opportunities for creating more inclusive and culturally responsive urban agriculture initiatives.
- Limitations should be moved before the Conclusion
Well noted! We have moved the limitation section before the conclusion section!
- Limitations
Recruitment through community organizations may have underrepresented or excluded recent or certain groups of immigrants. However, previous studies have adopted similar recruitment strategies and this approach enabled access to participants from 13 African countries, providing diverse perspectives. Second, while findings from Edmonton may not generalize to other Canadian cities, our socio-ecological model (SEM) offers an accessible template for examining similar dynamics in other cities. Third, although focusing only on Black African immigrants excluded other immigrant groups, this targeted approach revealed important insights about sociocultural, and structural barriers faced by this growing yet understudied population.
